# Febrile Seizure Causes Deficit in Social Novelty, Gliosis, and Proinflammatory Cytokine Response in the Hippocampal CA2 Region in Rats

**DOI:** 10.3390/cells12202446

**Published:** 2023-10-13

**Authors:** Yeon Hee Yu, Seong-Wook Kim, Hyuna Im, Yu Ran Lee, Gun Woo Kim, Seongho Ryu, Dae-Kyoon Park, Duk-Soo Kim

**Affiliations:** 1Department of Anatomy, College of Medicine, Soonchunhyang University, Cheonan-si 31151, Republic of Korea; yyh0220@sch.ac.kr (Y.H.Y.); hyuna990123@sch.ac.kr (H.I.); kbk1999@sch.ac.kr (Y.R.L.); 20227012@sch.ac.kr (G.W.K.); 2Graduate School of New Drug Discovery & Development, Chungnam National University, 99 Daehak-ro, Yuseong-gu, Daejeon 34134, Republic of Korea; seongwook0205@gmail.com; 3Soonchunhyang Institute of Med-Bio Science (SIMS), Soonchunhyang University, Cheonan-si 31151, Republic of Korea; ryu@sch.ac.kr; 4Department of Pathology, College of Medicine, Soonchunhyang University, Cheonan-si 31151, Republic of Korea

**Keywords:** febrile seizure, autism spectrum disorder, CA2 region, inflammation, gliosis

## Abstract

Febrile seizure (FS), which occurs as a response to fever, is the most common seizure that occurs in infants and young children. FS is usually accompanied by diverse neuropsychiatric symptoms, including impaired social behaviors; however, research on neuropsychiatric disorders and hippocampal inflammatory changes following febrile seizure occurrences is very limited. Here, we provide evidence linking FS occurrence with ASD pathogenesis in rats. We developed an FS juvenile rats model and found ASD-like abnormal behaviors including deficits in social novelty, repetitive behaviors, and hyperlocomotion. In addition, FS model juvenile rats showed enhanced levels of gliosis and inflammation in the hippocampal CA2 region and cerebellum. Furthermore, abnormal levels of social and repetitive behaviors persisted in adults FS model rats. These findings suggest that the inflammatory response triggered by febrile seizures in young children could potentially serve as a mediator of social cognitive impairments.

## 1. Introduction

Hyperthermic seizures can cause serious effects such as acute hippocampal neuronal injury and inflammatory response [1,2]. In our previous research, in which we established a febrile seizure model, we also confirmed the axonal reorganization processes following neuronal damage resulting from febrile seizures [3]. Other studies in rodents also showed that seizures in infancy or FSs due to genetic mutations cause hyperactivity, impaired social behavior, and cognitive memory [4,5]. Childhood epilepsy is often associated with neuropsychiatric disorders such as depression, anxiety, autism spectrum disorder, sleep disorder, attention deficit, cognitive disorder, and migraine [6].

According to a cohort analysis in Sweden, the prevalence of Early Symptomatic Syndromes Eliciting Neurodevelopmental Clinical Examinations (ESSENCE), including autism spectrum disorders, was significantly higher in children who experienced febrile seizures compared to those who did not experience seizures [7]. Animals with neonatal insult that results in persistent impairment, especially of social interaction, are considered useful for the study of neurodevelopmental disorders, such as schizophrenia and autism-spectrum disorder (ASD) [8]. ASD is a developmental neuropsychiatric disorder characterized by symptoms such as impaired social interaction, stereotyped behavior, and delayed language development [9]. Notably, attention deficit hyperactivity disorder (ADHD) and ASD are common mental disorders in children with epilepsy and FSs [10]. In addition, although hyperactivity disorder and ASD do not appear together in all autistic children, hyperactivity is one of the most common symptoms in most children with ASD [11]. In our previous study, we also observed hyperlocomotor activity in the FS animal model [12].

Upregulation of interleukin 6 (IL-6), which acts as a proinflammatory cytokine, and anti-inflammatory myokine is involved in physiological brain development, and, interestingly, plays opposing roles in several neurological disorders such as Alzheimer’s disease, multiple sclerosis, and excitotoxicity [13]. Prenatal IL-6 exposure in rats induced inflammatory neurodegeneration in the hippocampus and impaired spatial learning in adulthood [14]. Thus, IL-6 signaling has been proposed as a key mechanism of maternal immune activation that may be associated with ASD [15]. Inflammation triggers the release of many cytokines and signaling molecules, some of which have been found to predispose to FS and alter long-term synaptic plasticity in the hippocampus [16,17].

Dysfunctional hippocampal CA2 pyramidal neurons caused impairment of social memory in genetic and ASD models [18,19]. Especially, social memory deficits in adolescent rats exposed to neonatal status epilepticus (SE) episodes may be associated with the inhibition of vasopressin 1b receptor signaling in CA2 pyramidal cells and impaired long-term synaptic plasticity in the hippocampus [8,20].

While hyperthermic seizures have been widely associated with hippocampal neuronal injury, to the best of our knowledge, there are limited data on neuronal changes in hippocampal CA2 following a hyperthermic convulsion. Therefore, we aim to verify the association between inflammation in the hippocampal CA2 region caused by febrile seizures (FS) and deficits in social memory. 

## 2. Materials and Methods

### 2.1. Animals

Progeny of Sprague-Dawley rats were obtained from Experimental Animal Center, Soonchunhyang University (Cheonan, Republic of Korea). All animals were provided with a commercial diet and water and fed ad libitum under controlled temperature, humidity, and lighting conditions (light/dark cycle 12:12, 22 ± 2 °C, and 55 ± 5%). All animal protocols were approved by the Administrative Panel on Laboratory Animal Care of Soonchunhyang University (permit No. SCH22-0145). All efforts were made to minimize the number of animals used and their suffering.

### 2.2. Febrile Seizure Induction

A hyperthermic seizure was induced in rat pups (11 days postnatal) as previously described [3,12]. After punch-marking their ears, the pups were warmed in a plastic chamber measuring 10 cm × 13 cm at the base and 12 cm in height. The chamber floor was covered with paper towels. A 175 W mercury vapor lamp was held 3 cm above the chamber. Core temperatures were measured using an ear thermoprobe before inducing hyperthermic seizure and every 5 min during seizure induction. All rats showed generalized seizures at 5–10 min after hyperthermic seizure induction when the core temperature was 41–43 °C. Hyperthermic seizures were maintained for 40 min after generalized seizure onset because FS lasts for at least 15 min with more than one seizure in a day, resulting in transient neuronal injury and epilepsy [21,22]. During maintenance of hyperthermic seizures, rats showed multiple generalized seizures (stages 4 and 5) [23]. After the hyperthermic seizure, the rats were moved to a cool surface and returned to their mothers. Siblings placed in a chamber at room temperature were used as controls. To confirm FS inductions, we assessed the rats for recurrent seizures at intervals of approximately 4 h each day in the vivariums for general behavior (Racin scale criteria, 2.3 ± 0.07). In addition, we double cross-checked by performing LFPs monitoring to examine altered hippocampal oscillations caused by FS induction [12,24].

### 2.3. Behavioral Tests

All rats were tested for behavioral traits at 3 weeks (FS 3-weeks) and 12 weeks after FS induction (FS 12-weeks), respectively. Behavioral test results were recorded and analyzed using a PC-based video behavior analysis system with automated tracking software Noldus EthoVision XT 14 (Noldus, Leesburg, VA, USA).

#### 2.3.1. Social Recognition Test: Sociability and Social Novelty

The social recognition paradigm was adapted as described previously [24]. In brief, the apparatus was a rectangular acrylic box (FS 3-weeks: 60 cm length × 40 cm width × 20 cm height, FS 12-weeks: 120 cm length × 80 cm width × 40 cm height), divided into three areas of equal width and containing 2 wire cages that allowed the animals to explore one another but prevented aggressive interactions. On day 1, habituation was conducted for 10 min. On day 2, to assess sociability, a first novel rat (stranger 1) that had no prior contact with the subject rat was placed in the wire cage in one of the side chambers. The other compartment was an empty wire cage. Investigation was carried out during a 10 min test in which the subject rat sniffed an unfamiliar rat or climbed each wire cage. On day 3, a 10 min preference social novelty test was conducted in which a first novel rat (stranger 1) was placed in the left compartment and another novel rat (stranger 2) was placed in the other compartment. A rat expressing a normal preference for social novelty will spend a greater amount of time sniffing and climbing the compartment with a novel unfamiliar rat (stranger 2) compared with the compartment containing the first novel rat (stranger 1).

#### 2.3.2. Marble-Burying Test

Changes in repetitive behavior were evaluated through the marble-burying test in line with previously described methods [25]. The apparatus was a clean standard polycarbonate cage (FS 3-weeks: 40 cm length × 25 cm width × 17 cm height, FS 12-weeks: 50 cm length × 30 cm width × 20 cm height) covered with bedding measuring a height of 5 cm. A total of 20 glass marbles (FS 3-weeks: 15 mm, FS 12-weeks: 25 mm) were arranged in the cage in a four by five array. The test rat was placed in the apparatus and allowed to bury marbles freely for 30 min. Marbles were considered buried when more than two-thirds of their volume was covered by the bedding.

#### 2.3.3. Open-Field Test

The open-field arena was designed as 40 cm × 40 cm × 40 cm under diffuse lighting. Rats were placed in the center zone of the open-field arena and were allowed to explore freely for 30 min. Total distances moved were measured to evaluate locomotor activity [26].

### 2.4. Tissue Processing

Animals were anesthetized (urethane 1.5 g/kg, i.p.) and perfused transcardially with phosphate-buffered saline (PBS) followed by 4% paraformaldehyde in 0.1 M PB. The brains were extracted, post-fixed in the same fixative for 4 h, and rinsed in PB containing 30% sucrose at 4 °C for 2 days. Thereafter, the tissues were cryocut using a microtome at 30 µm. For the stereological study, consecutive sections throughout the entire hippocampus were used in some animals for immunohistochemistry and immunofluorescence experiments.

### 2.5. Immunohistochemistry and Immunofluorescence

Sections were incubated with each primary antibody in PBS containing 0.3% Triton X-100 overnight at 4 °C: rabbit anti-interleukin-6 (IL-6) IgG (diluted 1:500; Abcam, Trumpington, Cambridge, UK), rabbit anti-ionized calcium-binding adapter molecule 1 (Iba-1) IgG (diluted 1:200; Wako, Chuo-ku, Osaka, Japan), and mouse anti-glial fibrillary acidic protein (GFAP) IgG (diluted 1:1000; Millipore, Burlington, MA, USA). The sections were washed with PBS three times for 10 min, incubated sequentially in biotinylated goat anti-rabbit IgG and goat anti-mouse IgG (Vector laboratories, Burlingame, CA, USA), diluted 1:250 in the same solution as the primary antiserum and Avidin-Biotin Complex (Vector laboratories, Burlingame, CA, USA) in TBS-T. Between the incubations, the tissues were washed with PBS three times for 10 min each. The sections were visualized with 3,3′-diaminobenzidine in D.W. and mounted on gelatin-coated slides. The immunoreactions were observed using a DMRB microscope (Leica, Wetzlar, Hesse, Germany), and images were captured using a model DP72 digital camera and DP2-BSW microscope digital camera software (Olympus, Shinjuku, Tokyo, Japan). Double immunofluorescence staining was performed to identify the morphological changes induced by a hyperthermic seizure in the same hippocampal and cerebellum tissue. The brain tissues were incubated overnight at room temperature in a mixture of rabbit anti-IL-6 IgG (diluted 1:250)/mouse anti-striatal enriched tyrosine phosphatase (STEP, CA2 marker) IgG (diluted 1:500; Cell Signaling Technology, Danvers, MA, USA), goat anti-IL-6 IgG (diluted 1:300; Santa Cruz, Dallas, TX, USA)/rabbit anti-Iba-1 IgG (diluted 1:100), and mouse anti-GFAP IgG (diluted 1:500)/rabbit anti-IL-6 IgG (diluted 1:250). After washing with PBS three times for 10 min, the slices were incubated in a mixture of Cy2- and Cy3-conjugated secondary antisera (diluted 1:200, Sigma, St. Louis, MO, USA) for 1 h at room temperature. Then, these slices were incubated in DAPI (diluted 1:500; Invitrogen, Middlesex Country, MA, USA) as a counterstaining for 15 min at room temperature. After washing with PBS, the slices were placed on a slide and mounted with DPX (Sigma). All the images were captured using a model Fluoview FV10i and FV10i software (Olympus, Shinjuku, Tokyo, Japan).

### 2.6. Quantification of Data and Statistical Analysis

Quantification of data and statistical analysis of each data were performed as described in the previous study with some modifications [3,12]. Optical fractionation was used to estimate the cell numbers. The technique combines optical dissection counting and fractionator sampling, and is a stereological method based on a well-designed, systematic random sampling method. By definition, the approach yields unbiased estimates of a population. Samples of deep tissue were used (optical dissector height (h) was 30 µm). Statistical analyses were performed using GraphPad Prism 7 (GraphPad Software, Boston, MA, USA). All data were analyzed using Student’s t-test to determine statistical significance. A *p*-value of <0.01 or <0.001 was considered statistically significant.

## 3. Results

### 3.1. FS Model Rats Exhibit Deficits of Social Novelty, Increased Repetitive Behaviors, and Hyperlocomotion

We first investigated the social behaviors of rats using a three-chamber task. In the sociability test (Figure 1A-1), as measured by exploration time, the FS 3-weeks rats (Figure 1A-3) and wild-type littermates (Figure 1A-2) preferred to explore the first novel rat (stranger 1) over an empty cage (WT, *p* < 0.001; FS 3-weeks, *p* < 0.001; Student’s *t*-test), indicating that sociability was intact in FS 3-weeks rats. In the social novelty task (Figure 1B-1), when the empty cage was filled with another novel rat (stranger 2), wild-type rats preferred to explore stranger 2 over stranger 1 (*p* < 0.01; Student’s *t*-test; Figure 1B-2); however, FS 3-weeks rats did not show a preference for the stranger 2, indicating deficits of social novelty recognition (*p* = 0.46; Student’s t-test; Figure 1B-3). We further observed that the FS 3-weeks rat model exhibited increased repetitive behavior, a behavioral measure of ASD-like endophenotypes (Figure 2A-1), compared to the wild-type rats in the marble-burying task (*p* < 0.001; Student’s *t*-test; Figure 2A-2). To verify whether FS occurrence affects locomotor activities, an open-field test was performed with the FS 3-weeks rats and wild-type littermates (Figure 3A-1). FS 3-weeks rats showed longer distance moved than wild-type rats (*p* < 0.001; Student’s *t*-test; Figure 3A-2). Similar to the FS 3-weeks rat models, we observed social novelty impairment and repetitive behavior in the FS 12-weeks rat models (Figure 1C-1,C-2,D-1,D-2 and 2B-1,B-2). 

### 3.2. IL-6 Overexpression in the Hippocampal CA2 Region and Cerebellum of FS Model Rats

To verify whether hyperthermic seizure affects inflammation induction and immunoreactivity of Purkinje cell populations in the cerebellum, we performed IL-6 immunostaining (Figure 4A2–C3). In the FS 3-weeks rats, quantitation of the relative densities and an average number of IL-6-positive Purkinje cell populations in the cerebellum were analogously enhanced compared with the wild-type rats (*p* < 0.01; Student’s *t*-test; Figure 4D and *p* < 0.001; Student’s *t*-test; Figure 4E).

To further verify whether the inflammatory damages in the hippocampal CA2 region following hyperthermic seizure were related to the IL-6 expressions, we evaluated the levels of IL-6, Iba-1, and GFAP expression in the hippocampus through immunohistochemical analysis (Figure 5, Figure 6 and Figure 7). In FS 3-weeks rats, the IL-6 immunoreactive proinflammatory cytokine was markedly enhanced in the hippocampus compared with the wild-type rats (Figure 5A2,A3). In FS 12-weeks rats, the expression of IL-6 was not changed. Similarly, quantitation revealed increased relative densities of IL-6-positive neurons in FS 3-weeks rats compared with the wild-type rats (*p* < 0.001; Student’s *t*-test; Figure 5D). To identify the IL-6 expressional alterations in the hippocampal CA2 region of FS 3-weeks rats, we performed double immunofluorescent labeling with IL-6 and STEP (Figure 5B1–C3). In FS 3-weeks rats, IL-6-positive neurons were significantly increased in the hippocampal CA2 region compared with the wild-type rats (*p* < 0.001; Student’s *t*-test; Figure 5E). The Western blot experiments also showed similar results with immunohistochemical data (*p* < 0.001; Student’s *t*-test; Figure 5F,G). These results indicated that there was an increase in the IL-6-associated inflammatory response in the cerebellum and hippocampal CA2 region in juvenile rats after FS.

### 3.3. Gliosis in the Hippocampal CA2 Region Following Hyperthermic Convulsions

To confirm the association between glial activation and inflammation following FS, we performed double immunofluorescent labeling of IL-6/Iba-1 and GFAP/IL-6 (Figure 6B1–C4 and Figure 7B1–C4). In FS 3-weeks rats, the Iba-1 immunoreactive microglia with hypertrophic/elongated morphologies were observed in the hippocampal CA2 region; however, microglia with abnormal cellular morphologies were not found in the CA2 of wild-type rats (Figure 6A2,A3). Quantitation of the relative densities of Iba-1-positive glia revealed results that were similar to those of Iba-1 immunoreactivity (*p* < 0.001; Student’s *t*-test; Figure 6D). In FS 3-weeks rats, double immunofluorescent labeling glia with IL-6 and Iba-1 was significantly increased in the hippocampal CA2 region compared with wild-type rats (*p* < 0.001; Student’s *t*-test; arrows; Figure 6B1–C4,E). FS 3-weeks rats also had increased hypertrophic astroglial cells for astrogliosis as compared to wild-type rats (Figure 7A2,A3). Quantitation of the relative densities of GFAP-positive astrocytes revealed results that were similar to those of GFAP-immunoreactivity (*p* < 0.001; Student’s *t*-test; Figure 7D). In FS 3-weeks rats, double immunofluorescent labeling glia with GFAP and IL-6 was significantly increased in the hippocampal CA2 region compared with wild-type rats (*p* < 0.001; Student’s *t*-test; arrows; Figure 7B1–C4,E). These findings show that IL-6 expression and neuroinflammatory response caused by increased IL-6 in microglia and astrocytes were increased in the hippocampal CA2 region neurons.

## 4. Discussion

In previous studies, we provided evidence of not only an increase in anxiety and depression following febrile seizures but also hyperactivity of the locomotor phenotype in FS 3-weeks rats [3]. Changes in these specific phenotypes may indicate the possibility of additional neurodevelopmental disorders, such as cognitive and social behavioral impairments, occurring in the developmental stages following febrile seizures [4,5,6].

### 4.1. Disordered Social Novelty and Increased Repetitive Behaviors after Hyperthermic Convulsions

The core features of autistic disorder are the impairment in reciprocal social interaction, language deficit, stereotyped and repetitive behaviors, and a restricted range of interests and activities [27]. To directly assess the alteration of social behavior during developmental stages following a hyperthermic seizure, we compared the sociability performance between the FS 3-weeks rats and wild-type rats. Social novelty results showed that while wild-type rats spent more time exploring the second novel rat than the first novel rat, FS 3-weeks rats showed no preference for the novel rat. This indicates declined social novelty in the developmental stage after FS. Notably, social cognition tests are short-term working memory tests, a target mechanism for evaluating cognitive deficits associated with neuropsychiatric disorders such as Alzheimer’s disease, schizophrenia, Parkinson’s disease, and ASD [28]. Deficits in social cognition are evident in a variety of neurological, neurodegenerative, and psychiatric disorders [29], and are also common in patients with epilepsy, and the earlier the onset of seizures, the more pronounced these deficits [30]. Although research on social cognition in epilepsy is limited and insufficient, previous studies have demonstrated that early experience of prolonged FS leads to memory deficits in adult rats [31], and status epilepticus models of development phase show impaired social memory in adulthood [8]. We also observed social novelty impairments that persisted in adulthood, 12 weeks after FS. Repetitive restrictive behavior (RRB) is a nonspecific symptom commonly observed in neurodevelopmental and neuropsychiatric disorders [32]. RRBs are nonsensical patterns of behavior that interfere with normal behavior and have been identified by the American Psychiatric Association as a core symptom of ASD [33]. Following up on repetitive behaviors throughout development is an important factor in determining whether children are abnormal, and in children with ASD, RRB is maintained or aggravated over time [34]. In our results, the persistence of social novelty deficits and increased repetitive behaviors in FS 12-weeks rats may also support the potential outcomes of enduring neurodevelopmental and neuropsychiatric disorders following febrile seizures.

### 4.2. Hyperkinesia-like Personality Induced by Locomotor Deficits following FS

Similar to our previous research findings [12], the results of this current study have also demonstrated that hyperkinesia-like phenotype, a pattern of markedly enhanced locomotor activity in a novel environment, was demonstrated in 3-weeks rats after a hyperthermic seizure. Hyperactivity, impulsivity, aggression, self-injury, and irritability are disruptive behaviors that frequently accompany ASD [35]. Hyperkinesia was found in 72% of patients with ASD, and excessive exercise accompanies impulsivity and attention deficits over time [36]. ADHD is one of the most common mental disorders in children, and ADHD and ASD have common genetic factors [11,37]. Furthermore, altered dopamine-mediated synaptic potentiation during development is linked to structural and functional deficits in ADHD [38]. The immature dentate gyrus is identified in the endophenotype brain shared by several psychiatric disorders, and maturation of neurons and synaptic plasticity in the hippocampal circuit are crucial for working memory deficit and hyperlocomotor activity [39]. Seizures induced by FS caused significant synaptic reorganization in the immature dentate gyrus [3]. Thus, a synaptic reorganization that occurred at the developmental stage after FS may be accompanied by hyperkinesia-like behavior similar to ASD or ADHD. However, to confirm this hypothesis, further studies are needed.

### 4.3. Correlation between the Social Recognition Disorders and the Altered Functionalities of Hippocampal CA2 Neuronal Populations

In previous studies, increased expression of IL-6 in the cerebellum of autistic patients was suggested to stimulate excitatory synapse formation in granule cells, acting as a pathogenesis of ASD [40]. The cerebellum is involved in sensorimotor processing and motor control [41], and dysfunction of cerebellar Purkinje cell (PC) may be associated with RRBs [42]. Therefore, in our results, increased IL-6 in cerebellar PC cells may accompany RRB due to functional impairment of the cerebellum.

The CA2 region of the hippocampus was recently reported to play an important role in encoding declarative and social memories [18,24]. The impairment of the CA2 region and subsequent damage of social recognition provides evidence for the crucial role the CA2 plays in social cognition [18]. Therefore, we examined IL-6 immunoreactivity in the CA2 region of the hippocampus to investigate the association between the occurrence of social recognition disorders due to altered IL-6 expression and altered inflammatory responses. Our results showed that the immunoreactivity for IL-6 expression was increased in the hippocampal CA2 region of FS 3-weeks rats compared to that of wild-type rats. Inhibition of IL-6 trans-signaling in the ASD model increased synaptoneurosome-induced glutamate release from the cerebral cortex and alleviated abnormal social behavior [43]. This suggests that inhibition of excessive production of IL-6 in the CA2 region of the hippocampus following FS may be effective in treating ASD-like behavioral disorders. 

Reactive microglia release many cytokines and chemokines, including IL-1, IL-2, IL-6, tumor necrosis factor alpha (TNF-α), and interferon gamma (IFN-γ), that promote the neuroinflammation response, including via stimulating astrocyte activity [44]. On the other hand, glutamate uptake by excitatory amino acid transporters in astrocytes plays an important role in maintaining extracellular glutamate levels and modulating the activity of the surrounding synapses [45]. Reactive astrocytes release ATP to induce activation of microglia [46], and microglia and astrocytes intensify the inflammatory response through bidirectional communication [47]. In ASD brain tissue, microglia adjacent to neurons not only have a neuroprotective function but are also involved in excessive degradation of normal neurons, leading to synaptic dysfunction [48]. Therefore, in this study, the increased expression of microglia and astrocytes together with IL-6 after FS suggest an increase in the neuroinflammatory response in the CA2 region and subsequent to neuronal synaptic dysfunction.

## 5. Conclusions

Together, these findings may suggest that increased IL-6 due to the inflammatory response is an important mediator in abnormal social interactions. Furthermore, the observation of enhanced local neuroinflammation in the CA2 region and cerebellum, along with autism-like behavior following FS, is intriguing. However, additional studies are needed to confirm whether these factors cause ASD-like personality by genetic mutation after FS.

## Figures and Tables

**Figure 1 cells-12-02446-f001:**
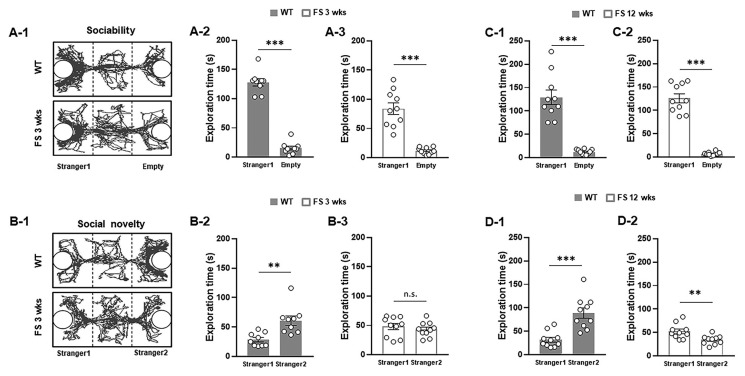
Impaired social novelty of FS model rats. Representative movement traces of FS 3-weeks rats and wild-type littermates in the three-chamber test (**A-1**,**B-1**). In the sociability task, both FS rats and wild-type rats preferred to explore the first novel rat (stranger 1) over an empty cage (**A-2**,**A-3**). In the social novelty task, when the empty cage was occupied by another novel rat (stranger 2), wild-type rats preferred to explore the stranger 2 over stranger 1; however, FS model rats did not show a preference for stranger 1 (**B-2**,**B-3**). The FS adult rats revealed normal sociability tasks (**C-1**,**C-2**) and preferred to explore stranger 1 over stranger 2 (**D-1**,**D-2**). Data are presented as means ± standard errors of the mean. ** *p* < 0.01, *** *p* < 0.001, two-tailed *t*-test (WT, *n* = 10; FS 3-weeks, *n* = 10; WT 12-weeks, *n* = 10; FS 12-weeks, *n* = 10).

**Figure 2 cells-12-02446-f002:**
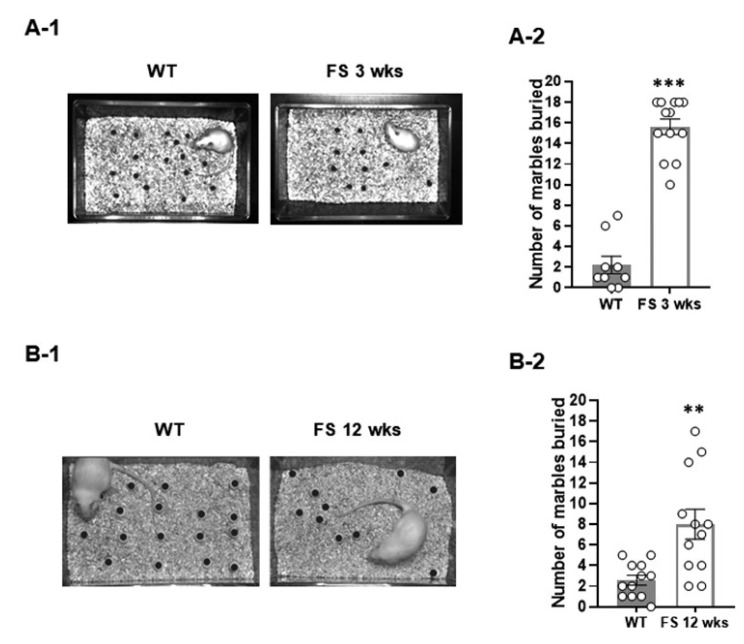
Increased repetitive behavior of FS rats. Repetitive behavior of FS 3-weeks rats in the marble-burying test (**A-1**). FS 3-weeks rats showed an increased level of marble-burying behavior compared to wild-type rats (**A-2**). Repetitive behavior of FS 12-weeks rats in the marble-burying test (**B-1**). FS 12-weeks rats showed increased levels of marble-burying behavior compared to wild-type rats (**B-2**). Data are presented as means ± standard errors of the mean. ** *p* < 0.01, *** *p* < 0.001, two-tailed *t*-test (WT 3-weeks, *n* = 10; FS 3-weeks, *n* = 10; WT 12-weeks, *n* = 12; FS 12-weeks, *n* = 12).

**Figure 3 cells-12-02446-f003:**
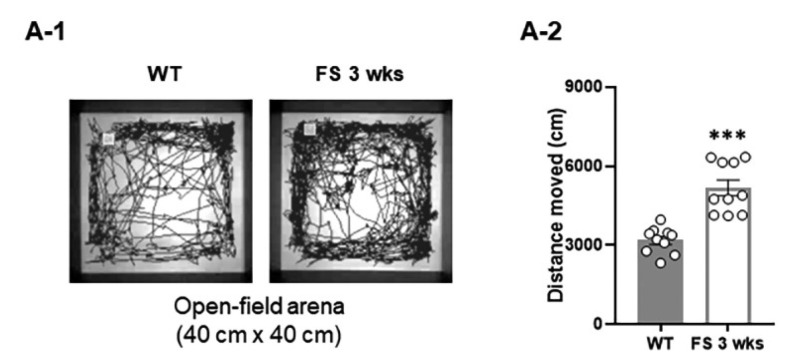
Hyperlocomotion of FS rats. Representative movement traces of FS 3-weeks rats and wild-type rats in the open-field test (**A-1**). Locomotor activity of FS 3-weeks rats was increased compared to wild-type rats (**A-2**). Data are presented as means ± standard errors of the mean. *** *p* < 0.001, two-tailed *t*-test (WT 3-weeks, *n* = 10; FS 3-weeks, *n* = 10).

**Figure 4 cells-12-02446-f004:**
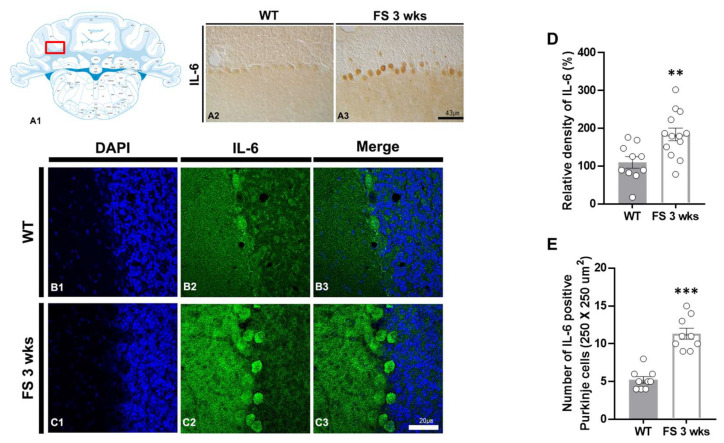
IL-6 upregulation in the cerebellum of FS rats. Diagram of the rat cerebellum (**A1**). The number of IL-6-positive Purkinje cells in the cerebellum of FS 3-weeks rats was elevated compared with wild-type rats (**A2**,**A3**). The IL-6 densitometric result was similar to the immunohistochemical data (**D**) (WT, *n* = 10; FS 3-weeks, *n* = 13). Immunofluorescence staining of IL-6 in the cerebellum of wild-type rats (**B1**–**B3**) and FS 3-weeks rats (**C1**–**C3**): DAPI (blue); IL-6 (green); merged images. The numbers of positive Purkinje cells for IL-6 were significantly higher compared to the wild-type rats (**E**). Data are presented as the mean ± standard errors of the mean. ** *p* < 0.01, *** *p* < 0.001, two-tailed *t*-test (WT, *n* = 9; FS 3-weeks, *n* = 9).

**Figure 5 cells-12-02446-f005:**
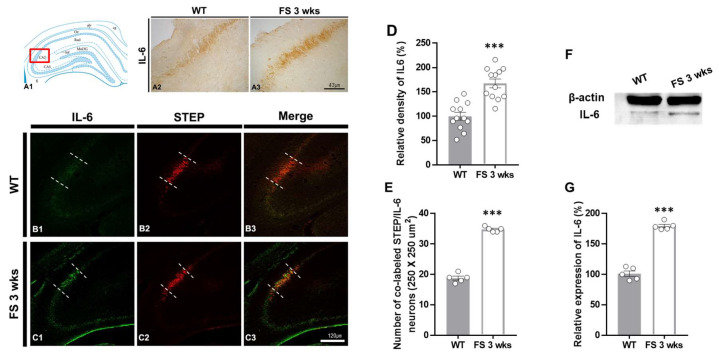
IL-6 upregulation in the CA2 hippocampal region of FS rats. Diagram of the rat hippocampus (**A1**). IL-6-positive neurons in the hippocampus of FS 3-weeks rats were elevated in the CA2 region compared with wild-type rats (**A2**,**A3**). The IL-6 densitometric result was similar to the immunohistochemical data (**D**) (WT, *n* = 12; FS 3-weeks, *n* = 12). Double labeling of IL-6 and STEP in the hippocampal CA2 region of the wild-type rats (**B1**–**B3**) and FS 3-weeks rats (**C1**–**C3**): IL-6 (green); STEP (red); merged images (yellow). The numbers of colocalized neuron IL-6 and STEP FS 3-weeks rats were significantly different from those of the wild-type rats (**E**). The immunoblot (**F**) and optical density analyses (**G**) showed more enhanced IL-6 in FS 3-weeks rats as compared to wild-type rats. Data are presented as the mean ± standard errors of the mean. *** *p* < 0.001, two-tailed *t*-test (WT, *n* = 5; FS 3-weeks, *n* = 5).

**Figure 6 cells-12-02446-f006:**
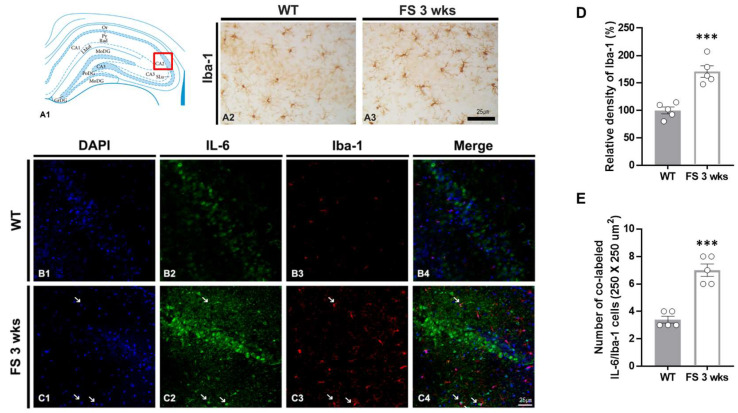
Microgliosis in the CA2 hippocampal region of the FS rats. Diagram of the rat hippocampus (**A1**). Iba-1 immunoreactive microglia in the FS 3-weeks rats showed hypertrophied cell body and hyper-ramified processes in the CA2 region unlike in wild-type rat (**A2**,**A3**). The Iba-1 densitometric results were similar to the immunohistochemical data (**D**) (WT, *n* = 5; FS 3-weeks, *n* = 5). Double labeling of IL-6 and Iba-1 in the hippocampal CA2 region of the wild-type rats (**B1**–**B4**) and FS 3-weeks rats (**C1**–**C4**): DAPI (blue); IL-6 (green); Iba-1 (red); merges (yellow). The numbers of double-label IL-6 and Iba-1 cells in the FS 3-weeks rats were significantly different from those of the wild-type rats (arrows, **C1**–**C4**,**E**). Data are presented as the mean ± standard errors of the mean. *** *p* < 0.001, two-tailed *t*-test (WT, *n* = 5; FS 3-weeks, *n* = 5).

**Figure 7 cells-12-02446-f007:**
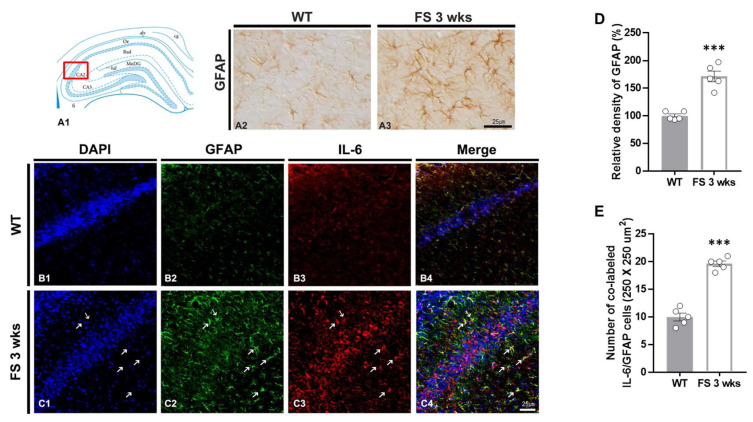
Astrogliosis in the CA2 hippocampal region of FS rats. Diagram of the rat hippocampus (**A1**). GFAP-immunoreactive astrocytes in the FS 3-weeks rats were hypertrophied cell body and significantly increased in the CA2 region compared with wild-type rats (**A2**,**A3**). The GFAP densitometric results were similar to the immunohistochemical data (**D**) (WT, *n* = 5; FS 3-weeks, *n* = 5). Double labeling of GFAP and IL-6 in the hippocampal CA2 region of wild-type rats (**B1**–**B4**) and FS 3-weeks rats (**C1**–**C4**): DAPI (blue); GFAP (green); IL-6 (red); merges (yellow). The numbers of double-labeled IL-6 and GFAP cell were significantly different from those of the wild-type rats (arrows **C1**–**C4**,**E**). Data are presented as the mean ± standard errors of the mean. *** *p* < 0.001, two-tailed *t*-test (WT, *n* = 5; FS 3-weeks, *n* = 5).

## Data Availability

Not applicable.

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
