# Peer review of "Febrile Seizure Causes Deficit in Social Novelty, Gliosis, and Proinflammatory Cytokine Response in the Hippocampal CA2 Region in Rats"

_cells, 2023, doi:10.3390/cells12202446_

Round 1

Reviewer 1 Report

Febrile seizures, which occur in response to fever, are common in infants and young children. Although febrile seizures are often accompanied by neuropsychiatric symptoms, such as impaired social behavior, their connection to autism spectrum disorder (ASD) is not fully understood. In this study, researchers developed a rat model of febrile seizures and found that the rats exhibited ASD-like behaviors, including deficits in social novelty, repetitive behaviors, and hyperlocomotion. The rats also showed increased levels of inflammation in the hippocampal CA2 region and cerebellum. These abnormal behaviors and inflammation persisted into adulthood. These findings suggest that febrile seizures in infants may be associated with the development of ASD-like characteristics. The researchers also investigated the role of IL-6, a pro-inflammatory cytokine, in this process. They found increased expression of IL-6 in the hippocampal CA2 region and cerebellum of the rats. Additionally, they observed the activation of microglia and astrocytes, which are involved in inflammatory responses. These results suggest that the increased inflammation and IL-6 expression may contribute to the development of ASD-like behaviors. Overall, this study provides evidence linking febrile seizures to ASD pathogenesis in a rat model and highlights the potential role of inflammation in this process. The group has a strong background in studying FS association with psychiatric disorders, depression, anxiety, autism spectrum disorders, attention deficits, and cognitive impairment. This study is well planned; material methods are correctly mentioned, and their results are appropriately discussed.

However, I have some minor comments and suggestions that I believe will further enhance the manuscript:

1.    Figure 3: In Figure 3, you present data on hyper-locomotion in FS 3-week-old rats compared to wild-type rats. It has come to my attention that similar data were published in a previous study by your group (Yu et al., Cells 2022, Fig.3; https://doi.org/10.3390/cells11203228). To avoid redundancy and to clarify the novelty of the findings in this manuscript, please explicitly state whether the data in Figure 3 are new or if they overlap with previously published data. If the data are the same or very similar, provide a brief explanation or reference to the previous study.

2.    IL-6 Expression at 12 Weeks: Your study investigates IL-6 overexpression in the hippocampal CA2 region and cerebellum of FS 3-week-old rats. It would be valuable to extend this analysis to FS 12-week-old rats to determine whether the high IL-6 expression persists or returns to normal. This longitudinal perspective would provide insights into the long-term effects of FS on IL-6 expression and help clarify whether increased inflammation is a sustained consequence of seizures or is specifically linked to ASD-like behaviors.

3.    Correlation Analysis: To establish a more direct link between seizure activity and IL-6 expression, consider adding a correlation graph that illustrates the relationship between hyperthermic seizures, generalized seizures, and IL-6 expression in the hippocampal CA2 region and cerebellum of FS rats. This analysis would enhance the comprehensibility of the results and their implications.

I appreciate the thoroughness of your study and believe that addressing these points will strengthen the manuscript's clarity and relevance.

Author Response

Reviewer 1

Comments and Suggestions for Authors

Febrile seizures, which occur in response to fever, are common in infants and young children. Although febrile seizures are often accompanied by neuropsychiatric symptoms, such as impaired social behavior, their connection to autism spectrum disorder (ASD) is not fully understood. In this study, researchers developed a rat model of febrile seizures and found that the rats exhibited ASD-like behaviors, including deficits in social novelty, repetitive behaviors, and hyperlocomotion. The rats also showed increased levels of inflammation in the hippocampal CA2 region and cerebellum. These abnormal behaviors and inflammation persisted into adulthood. These findings suggest that febrile seizures in infants may be associated with the development of ASD-like characteristics. The researchers also investigated the role of IL-6, a pro-inflammatory cytokine, in this process. They found increased expression of IL-6 in the hippocampal CA2 region and cerebellum of the rats. Additionally, they observed the activation of microglia and astrocytes, which are involved in inflammatory responses. These results suggest that the increased inflammation and IL-6 expression may contribute to the development of ASD-like behaviors. Overall, this study provides evidence linking febrile seizures to ASD pathogenesis in a rat model and highlights the potential role of inflammation in this process. The group has a strong background in studying FS association with psychiatric disorders, depression, anxiety, autism spectrum disorders, attention deficits, and cognitive impairment. This study is well planned; material methods are correctly mentioned, and their results are appropriately discussed.

However, I have some minor comments and suggestions that I believe will further enhance the manuscript:

  1. Figure 3: In Figure 3, you present data on hyper-locomotion in FS 3-week-old rats compared to wild-type rats. It has come to my attention that similar data were published in a previous study by your group (Yu et al., Cells 2022, Fig.3; https://doi.org/10.3390/cells11203228). To avoid redundancy and to clarify the novelty of the findings in this manuscript, please explicitly state whether the data in Figure 3 are new or if they overlap with previously published data. If the data are the same or very similar, provide a brief explanation or reference to the previous study.

Response: With respect to reviewer’s comments, we rewritten the discussion part of revised manuscript as below (Page 10 Line 354 - 357):

“Similar to our findings in previous research [12], the results of this current study have also demonstrated hyperkinesia-like phenotype, a pattern of markedly enhanced locomotor activity in a novel environment, was demonstrated in 3-weeks rats after a hyperthermic seizure.”

  1. IL-6 Expression at 12 Weeks: Your study investigates IL-6 overexpression in the hippocampal CA2 region and cerebellum of FS 3-week-old rats. It would be valuable to extend this analysis to FS 12-week-old rats to determine whether the high IL-6 expression persists or returns to normal. This longitudinal perspective would provide insights into the long-term effects of FS on IL-6 expression and help clarify whether increased inflammation is a sustained consequence of seizures or is specifically linked to ASD-like behaviors.

Response: Thank reviewer for important comments. Following reviewer’s suggestion, we provided evidences showing for IL-6 at 12 weeks after FS. After 12 weeks following FS induction, there was no IL-6 upregulation in the hippocampal CA2 region (figure below). Based on our previous research, it is speculated that the recurrent seizures occurring 12 weeks after FS induction may result from abnormal regulation of excitatory and inhibitory processes due to synaptic changes (Yu et al., 2017).

With respect to reviewer’s comments, we have added the following sentence to the results section. ‘In FS 12-weeks rats, the expression of IL-6 was not changed (data not shown)’ (Page 7 Line 241 - 242).

  1. Correlation Analysis: To establish a more direct link between seizure activity and IL-6 expression, consider adding a correlation graph that illustrates the relationship between hyperthermic seizures, generalized seizures, and IL-6 expression in the hippocampal CA2 region and cerebellum of FS rats. This analysis would enhance the comprehensibility of the results and their implications.

Response: The FS group used in the experiments consisted of rats that all exhibited hyperthermic seizures at the time of FS induction. Subsequently, in the febrile seizure-induced rats at 12 weeks, recurrent seizures occurred in approximately 71% of them (Yu et al., 2017). Similarly, social recognition deficits occurred in approximately 71% (n= 10/14) of the FS3wks rats, while the remaining 29% of rats did not show significant changes in IL-6 or glial expression.

I appreciate the thoroughness of your study and believe that addressing these points will strengthen the manuscript's clarity and relevance.

References

  1. Gillberg, C.; Lundström, S.; Fernell, E.; Nilsson, G.; Neville, B. Febrile Seizures and Epilepsy: Association With Autism and Other Neurodevelopmental Disorders in the Child and Adolescent Twin Study in Sweden. Pediatr. Neurol. 2017, 74, doi:10.1016/j.pediatrneurol.2017.05.027.
  2. Nilsson G.; Lundström S.; Fernell E.; Gillberg C.Neurodevelopmental problems in children with febrile seizures followed to young school age: A prospective longitudinal community-based study in Sweden. Acta Paediatr. 2022, 111, 586-592, doi: 10.1111/apa.16171.
  3. Yu, Y.H.; Lee, K.; Sin, D.S.; Park, K.H.; Park, D.K.; Kim, D.S. Altered Functional Efficacy of Hippocampal Interneuron during Epileptogenesis Following Febrile Seizures. Brain Res. Bull. 2017, 131, 25–38, doi:10.1016/j.brainresbull.2017.02.009.

Reviewer 2 Report

Interesting and well designed study with interesting results. 

However, the discussion includes many assumptions that cannot be made based on this study.

The initial statement that febrile seizures are typically associated with ADHD and ASD and variety of neuropsychiatric disorders is incorrect. Most children with simple FS have normal neurodevelopmental outcome. Yes, children with ASD have higher rates of epilepsy but there is no evidence that they have higher prevalance of FS. Most children with ASD often show signs of abnormal communication and social interactions before they have first seizures and many don't have febrile seizures at all.

Authors connection between FS and ASD cannot be made based on this study or any current evidence. There is also currently no evidence linking ASD to an inflammatory process.

The statement "Neonatal seizures induced by FS" should be re-written, it is unclear. Neonatal seizures occur before FS, and seizures in neonates even in the context of fever are not called FS.

The animal model seems to be more fitting with prolonged FS that simple febrile seizure.

The study design and study results are interesting and the authors should consider re-writing the manuscript in a more descriptive way. Such as a possible abnormal neurodevelopmental and neuropsychiatric outcome following prolonged FS rather than connecting the results directly with ASD and ADHD.

Author Response

Reviewer 2

Comments and Suggestions for Authors:

Interesting and well designed study with interesting results. 

However, the discussion includes many assumptions that cannot be made based on this study

  1. The initial statement that febrile seizures are typically associated with ADHD and ASD and variety of neuropsychiatric disorders is incorrect. Most children with simple FS have normal neurodevelopmental outcome. Yes, children with ASD have higher rates of epilepsy but there is no evidence that they have higher prevalance of FS. Most children with ASD often show signs of abnormal communication and social interactions before they have first seizures and many don't have febrile seizures at all.

Response: Thanks for the comments from the reviewer's opinions. As similarly mentioned above comments, previous clinical investigations into the association between neuropsychiatric disorders and febrile seizures are limited. However, according to Sweden's child cohort analysis, children with a history of febrile seizures exhibited neurodevelopmental disorders, and the prevalence of Early Symptomatic Syndromes Eliciting Neurodevelopmental Clinical Examinations (ESSENCE) was also found to be high (Gillberg et al., 2017, Nilsson et al., 2022). Indeed, these research teams have also mentioned the need for additional studies determine whether febrile seizures should be considered as an indicator of neurodevelopmental disorders (Nilsson et al., 2022). So, we hope to consider the above recent research findings.

However, with respect the reviewer's opinion we have added and re-wrote the below sentences in the introduction part (Page 1 Line 44 - 47).

“According to a cohort analysis in Sweden, the prevalence of Early Symptomatic Syndromes Eliciting Neurodevelopmental Clinical Examinations (ESSENCE), including autism spectrum disorders, was significantly higher in children who experienced febrile seizures compared to those who did not experience seizures [7].”

  1. Authors connection between FS and ASD cannot be made based on this study or any current evidence. There is also currently no evidence linking ASD to an inflammatory process.

Response: Following reviewer’s comments, we re-wrote / removed the contents suggesting the involvement of FS-ASD and ASD-inflammatory process based on current our findings in the revised manuscript.

  1. The statement "Neonatal seizures induced by FS" should be re-written, it is unclear. Neonatal seizures occur before FS, and seizures in neonates even in the context of fever are not called FS.

Response: With respect to reviewer’s comments, we have removed the word 'neonatal' from the sentence (Page 11 Line 370).

The animal model seems to be more fitting with prolonged FS that simple febrile seizure.

The study design and study results are interesting and the authors should consider re-writing the manuscript in a more descriptive way. Such as a possible abnormal neurodevelopmental and neuropsychiatric outcome following prolonged FS rather than connecting the results directly with ASD and ADHD.

References

  1. Gillberg, C.; Lundström, S.; Fernell, E.; Nilsson, G.; Neville, B. Febrile Seizures and Epilepsy: Association With Autism and Other Neurodevelopmental Disorders in the Child and Adolescent Twin Study in Sweden. Pediatr. Neurol. 2017, 74, doi:10.1016/j.pediatrneurol.2017.05.027.
  2. Nilsson G.; Lundström S.; Fernell E.; Gillberg C.Neurodevelopmental problems in children with febrile seizures followed to young school age: A prospective longitudinal community-based study in Sweden. Acta Paediatr. 2022, 111, 586-592, doi: 10.1111/apa.16171.
  3. Yu, Y.H.; Lee, K.; Sin, D.S.; Park, K.H.; Park, D.K.; Kim, D.S. Altered Functional Efficacy of Hippocampal Interneuron during Epileptogenesis Following Febrile Seizures. Brain Res. Bull. 2017, 131, 25–38, doi:10.1016/j.brainresbull.2017.02.009.

Reviewer 3 Report

The manuscript is clear and well written. The Methods and results sections are detailed and easy to follow; I have online few comments for the authors:

- Introduction: I suggest to ad few lines in order to explain the affidability of the rat modèl U.S. ed in this study.

- Discussion: section 4.3: I suggest to add some studies carried out in humans in order to underline the possibile link: the authors must read and cite the paper by Parisi P te al. Epilepsy Behav 2014; 32:72-5

Few typo mistakes.

Author Response

Reviewer 3

Comments and Suggestions for Authors:

The manuscript is clear and well written. The Methods and results sections are detailed and easy to follow; I have online few comments for the authors:

  1. Introduction: I suggest to ad few lines in order to explain the affidability of the rat modèl U.S. ed in this study.

Response: With respect to reviewer’s comments, we have included the following content into the introduction section of revised manuscript (Page 1 Line 37 - 39).

“In our previous research, in which we established a febrile seizure rat model, we also confirmed the reorganization processes following neuronal damage resulting from febrile seizures [3].”

  1. Discussion: section 4.3: I suggest to add some studies carried out in humans in order to underline the possibile link: the authors must read and cite the paper by Parisi P te al. Epilepsy Behav 2014; 32:72-5

Response: Following the reviewer's comment, we cited the paper by Parisi P et al., Epilepsy Behav 2014; 32:72-5 (Page 11 Line 365 - 367).

“Furthermore, altered dopamine-mediated synaptic potentiation during development is linked to structural and functional deficits in ADHD [38].“

Comments on the Quality of English Language: Few typo mistakes.

References

  1. Gillberg, C.; Lundström, S.; Fernell, E.; Nilsson, G.; Neville, B. Febrile Seizures and Epilepsy: Association With Autism and Other Neurodevelopmental Disorders in the Child and Adolescent Twin Study in Sweden. Pediatr. Neurol. 2017, 74, doi:10.1016/j.pediatrneurol.2017.05.027.
  2. Nilsson G.; Lundström S.; Fernell E.; Gillberg C.Neurodevelopmental problems in children with febrile seizures followed to young school age: A prospective longitudinal community-based study in Sweden. Acta Paediatr. 2022, 111, 586-592, doi: 10.1111/apa.16171.
  3. Yu, Y.H.; Lee, K.; Sin, D.S.; Park, K.H.; Park, D.K.; Kim, D.S. Altered Functional Efficacy of Hippocampal Interneuron during Epileptogenesis Following Febrile Seizures. Brain Res. Bull. 2017, 131, 25–38, doi:10.1016/j.brainresbull.2017.02.009.

Round 2

Reviewer 2 Report

Thank you for making changes to the manuscript. 
It reads very well and brings up an interesting association between febrile seizures and neurodevelopmental impairment. 
The only suggestion I have for authors is to consider changing:

FS are “typically” associated with neuropsychiatric impairment (in introduction)

to “can be associated” since febrile seizures are commonly seen in children with normal neurological development. 

it would be interesting to see if age at febrile seizure matters or recurrent febrile seizures matter.